# Carbon-Saving Potential of Urban Parks in the Central Plains City: A High Spatial Resolution Study Using a Forest City, Shangqiu, China, as a Lens

Jianwei Gao [1,*,†], Haiting Han [2,3,†] and Shidong Ge [4,*]

1   School of Economics, Tianjin University of Commerce, Tianjin 300134, China
2   Department of Food and Resource Economics, Københavns Universitet, Rolighedsvej 23, 1953 Frederiksberg C, Denmark; hanht@pku.edu.cn
3   Research and Impact Assessment Division, International Fund for Agricultural Development, 00142 Rome, Italy
4   International Union Laboratory of Landscape Architecture, College of Landscape Architecture and Art, Henan Agricultural University, Zhengzhou 450002, China
*   Correspondence: gjw0923@tjcu.edu.cn (J.G.); shidongge@henau.edu.cn (S.G.)
†   These authors contributed equally to this work.

**Abstract:** This article investigates the potential for carbon reduction in urban parks in Shangqiu City using high-resolution remote sensing imagery. The aim is to guide modern urban carbon neutrality strategies. The carbon-saving potential is estimated based on the mitigation of the urban heat island effect by park greenery, which reduces energy consumption. Then, the sample parks were divided into different categories, and 16 landscape metrics were selected to analyze their relationship with carbon-saving potential and driving factors. We found that a total of 300.57 t $CO_2$ could be reduced in Shangqiu City parks, and on average, a park could reduce $2.55 \pm 0.31$ t $CO_2$ ($1.79 \pm 0.29$ t $CO_2$ ha$^{-1}$) per summer day. The significant effect of landscape patterns on park carbon-saving differs between park categories, which means that park carbon-saving enhancement strategies need to be different for different park categories. Meanwhile, this study implies that the landscape pattern can be designed to enhance the carbon-saving potential of urban parks, which can play a great role in promoting the process of carbon neutrality and mitigating climate change in China.

**Keywords:** urban park; carbon-saving potential; high spatial resolution; Shangqiu





## 1. Introduction

Changes in urban land cover have caused dramatic alterations in local urban microclimates, leading to a series of ecological issues. As urbanization continues to accelerate, extreme weather events and the urban heat island (UHI) effect have become hot topics among academics. The UHI effect refers to the phenomenon in which urban areas have significantly higher temperatures than adjacent rural areas. Although the theory was proposed early on, it remains a topic of great interest in the academic community, with many unresolved issues even after extensive research into its causes and mitigation strategies [1]. Some studies suggest that the UHI effect leads to the formation of local air circulation within cities, carrying air pollutants to higher altitudes and posing serious threats to human health [2]. In addition, a strong UHI effect can greatly impact the comfort of urban living conditions [3]. Especially in summer, the UHI effect is particularly pronounced, increasing the use of air conditioning and leading to higher energy consumption and carbon emissions. Therefore, reducing the UHI effect is a crucial component in achieving urban carbon neutrality.

Urban park green spaces have a significant impact on mitigating the urban heat island effect [4–7]. Many scholars have recognized the positive role of urban parks in offsetting this effect and creating "cool islands" within cities [8–10]. This conclusion is supported by

the results of studies in numerous countries, such as China [11,12], the USA [13], India [14], Portugal [15], and the UK [16]. There are several reasons for this. First, vegetation can provide shade, reducing direct solar radiation on the ground and decreasing the absorption of heat radiation by the ground [17–19]. Second, through the transpiration of plants and their low albedo, some of the solar radiation can be absorbed. Finally, blue–green infrastructure, including urban green spaces, can effectively reduce the surface temperature of adjacent areas by directing airflow within the city. Therefore, enhancing the "cool island" effect within urban areas is considered an effective way to mitigate the heat effect [20].

In the past, most studies on urban green spaces have focused on their direct carbon sequestration capacity, which increases as vegetation biomass increases [21]. For example, in Xi'an, urban green spaces have an annual fixed carbon amount of 0.70 t $CO_2$ ha$^{-1}$ [22], while in Beijing, Changchun, and Harbin, the average carbon densities are 28.6 t $CO_2$ ha$^{-1}$ [23], 31.9 t $CO_2$ ha$^{-1}$ [24], and 28.2 t $CO_2$ ha$^{-1}$ [25], respectively. However, other studies have pointed out that the role of vegetation in reducing energy consumption in urban areas and bringing about carbon emission reductions cannot be ignored [26,27]. An empirical study showed that park green spaces have great potential to reduce carbon emissions and could save $23.7 \pm 1.6$ t $CO_2$ in the Yangtze River Economic Belt region [26]. Therefore, it is critical to understand the potential carbon emission reduction capacity through reducing thermal environmental pressure. However, the potential carbon emission reduction capacity of urban green spaces has received limited attention, especially in central China. Shangqiu is the core city of China's Central Plains Economic Zone and one of the most promising cities in Henan Province, China. Since its establishment in 1997, Shangqiu has undergone rapid urbanization, with significant expansion of construction and transportation land use. From 2000 to 2015, the city's construction land use expanded rapidly, increasing by as much as 696.57 square kilometers, which accounted for 37.57% of the total construction land use in 2000. This rapid urbanization has led to significant changes in the city's landscape pattern, making it an ideal area to study the potential for carbon-saving in urban areas.

This study examines the potential for reducing carbon emissions in urban parks in Shangqiu, a city in Central China. Our objective is to provide guidance for urban development and construction from the perspective of urban ecology and environmental economics. Specifically, we aim to answer the following research questions: (1) What is the magnitude and spatial distribution of carbon-saving potential in Shangqiu's urban parks? (2) What are the landscape factors driving spatial heterogeneity in the carbon-saving potential across different urban parks in Shangqiu?

## 2. Materials and Methods

### 2.1. Study Area

Shangqiu is located in the eastern Henan province of China between longitudes 114°49′~116°39′ E and latitudes 33°43′~34°52′ N (Figure 1). It has a total area of 10,704 square kilometers and a population of 7.723 million. It is a national civilized city, national health city, national garden city, and national forest city, with tourist attractions like the ancient city of Shangqiu. The city's plains account for 99.2% of its total area, with only 0.8% being hilly areas. Shangqiu has a warm temperate semi-humid continental monsoon climate with four distinct seasons, ample sunshine, and abundant rainfall. The city has an annual average of 1944 h of sunshine, an average annual temperature of 14.6 °C, an average annual rainfall of 736.2 mm, and a frost-free period of about 211 days. In recent years, Shangqiu has gradually become a densely populated and economically developed large city in central China.

### 2.2. Methods

The research process is shown in Figure 2. Firstly, the park LST is extracted based on Landsat 8 remote sensing images, and the park's land use information is extracted based on GF-2 remote sensing images. Secondly, the park landscape pattern index was calculated based on the park's land use information. Finally, the influence of the landscape

pattern index on the carbon-saving potential of the park was explored by correlation and stepwise regression.

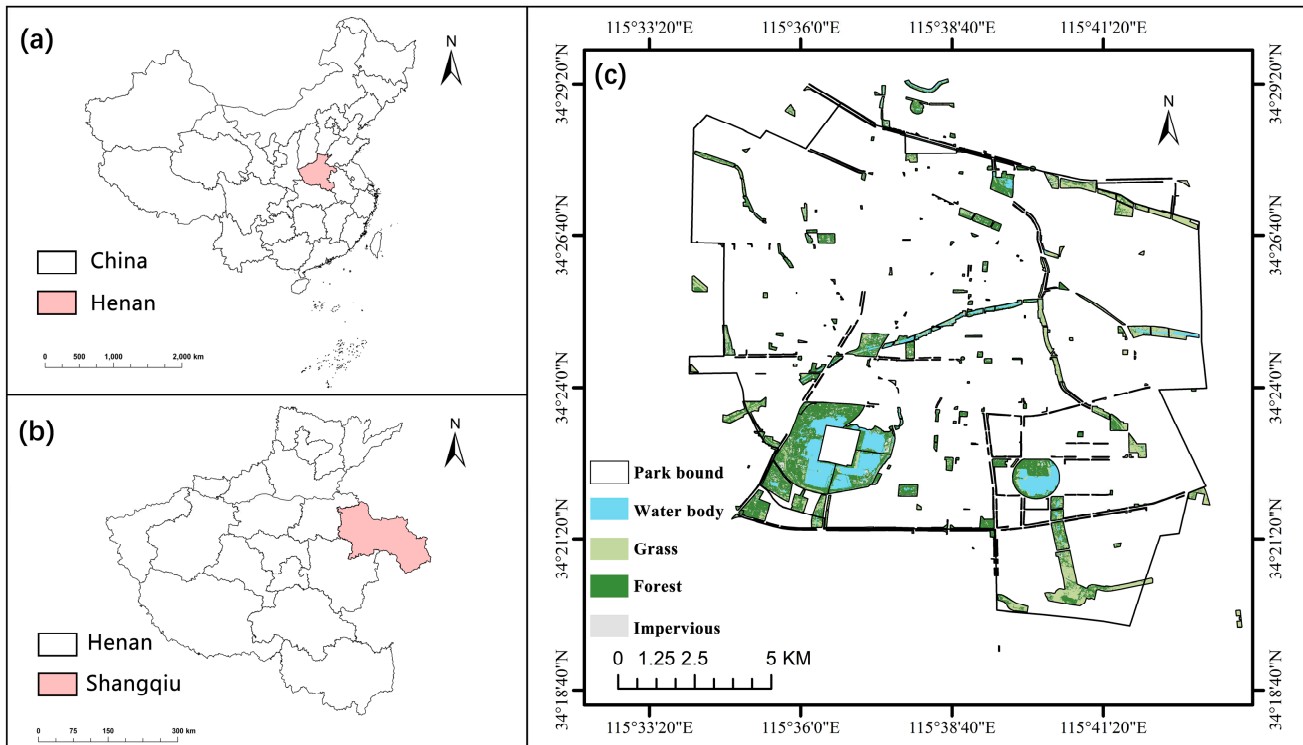

**Figure 1.** The study area's location with a background map displaying land cover data. (**a**): China, (**b**): Henan Province and Shangqiu City, and (**c**): main urban area of Shangqiu and land use data of 118 parks.

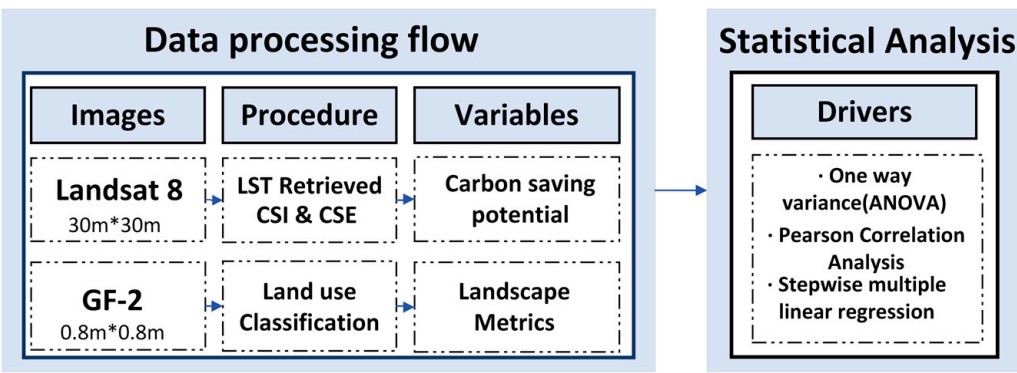

**Figure 2.** Flowchart of this study.

### 2.2.1. Remote-Sensed Urban Park

The Split-Window algorithm is a commonly used method in the field of land surface temperature (LST) remote sensing retrieval. It uses the ratio of reflectance in two spectral bands to remove atmospheric effects and obtain the land surface radiation temperature, which is then converted to LST using thermal radiation principles [28,29]. This algorithm applies to remote sensing image data of different resolutions and spectral bands and has the advantages of high accuracy and strong operability [30]. In this study, the original data were obtained from a Landsat 8 image data set with a spatial resolution of 30 m. All remote sensing images of cloudless and sunny days were selected from the summers of 2020 to 2021 (Table 1).

**Table 1.** Information on remotely sensed data for Shangqiu.

| Satellite | Path/Row | Peirod (Year-Month-Day) |
|---|---|---|
| Landsat | 122/36 | 2020-08-28 |
| Landsat | 122/36 | 2020-09-04 |
| Landsat | 122/36 | 2021-06-28 |
| Landsat | 122/36 | 2021-07-30 |
| Landsat | 122/36 | 2021-09-16 |
| GF-2 | —— | 2021-08-10 |

The radiative transfer equation used in the Split-Window algorithm for calculating LST is expressed as follows:

$$L_\lambda = \left[\varepsilon \cdot B(T_S) + (1-\varepsilon) \cdot L_\downarrow\right] \cdot \tau + L_\uparrow \tag{1}$$

where $L_\lambda$ represents the thermal radiation intensity of wavelength $\lambda$ received by the satellite sensor, $\varepsilon$ is the land surface emissivity, $B(T_S)$ is the radiation brightness received by a blackbody with temperature $T_S$, and its unit is $W \cdot m^{-2} \cdot sr^{-1} \cdot \mu m^{-1}$. $\tau$ is the atmospheric transmissivity, and $L_\uparrow$ and $L_\downarrow$ are the upwelling and downwelling atmospheric radiances obtained from NASA [31]. This equation, based on the principles of thermal radiation, considers the effects of the atmosphere on radiation and can be used to retrieve land surface temperature from satellite data. It is a crucial component of the Split-Window algorithm.

Later, we used the GF-2 image (Table 1) and the method of object-based image classification to interpret the land use in the study area. To improve the accuracy of interpretation, we optimized the samples based on auxiliary data and combined manual corrections by researchers to modify the parts with significant deviation, maximizing the accuracy of land use data. We divided the land use of 118 parks into four categories: forest (including trees and shrubs), grass, water bodies (including ponds, streams, etc.), and impervious surfaces (including buildings, squares, and other facilities) (Figure 1c).

Based on high-resolution remote sensing images, the 118 parks in Shangqiu City can be classified according to different features and classification standards (Table 2). Firstly, according to the nature of the park and the urban land classification standard (CJJ/T85-2017), parks can be classified into community parks, small parks, special parks, and comprehensive parks. Secondly, according to the size of the park's area, it can be divided into less than 2 ha parks, 2–5 ha parks, 5–10 ha parks, and more than 10 ha parks. Thirdly, the presence of water bodies within the park is also a classification criterion that can divide parks into parks with water and parks without water. Through these classification criteria, we can have a more comprehensive understanding of the distribution and characteristics of parks in Shangqiu City.

2.2.2. Landscape Metrics

Landscape indices are statistical measures used to describe and quantify landscape patterns. They are commonly used to analyze the impact of different land use types and landscape characteristics on ecosystems, including biodiversity, ecosystem services, and landscape functionality. In this study, 16 landscape pattern indices were selected (Table 3) based on the reference of the results of previous studies related to LST and landscape pattern indices in Shangqiu City and nearby cities [10,32,33]. These landscape pattern indices are calculated in Fragstats 4.2 based on the 8-unit neighborhood method [34].

These landscape indices are Number of Patches (NP), Patch Density (PD), Aggregation Index (AI), Contagion (CONTAG), Interspersion and Juxtaposition Index (IJI), Cohesion Index (COHESION), Splitting Index (SPLIT), Landscape shape index (LSI), Mean Patch Shape Index (SHAPE_MN), Mean Patch Area (PARA_MN), Mean Fragment Shape Index (FRAC_MN), Largest Patch Index (LPI), Mean Patch Area (AREA_MN), Shannon's Evenness Index (SHEI), Shannon's Diversity Index (SHDI), Percentage of Forest, Grass,

Impervious surfaces, and water (PLAND_Forest, PLAND_Grass, PLAND_Imper, and PLAND_Water).

**Table 2.** Park classification criteria and description.

| Classification Standards | Park Type | Description |
|---|---|---|
| Park character | Community park | The area is larger than 1 ha; the site is independent, with basic service facilities, mainly a service green space for the residents of a certain community to carry out daily leisure activities. |
| | Small park | Smaller areas or diverse shapes, independent sites, convenient for residents to access nearby, with certain recreational functions of the green space. |
| | Special park | A green space with specific content or form with corresponding service facilities; for example, zoos, botanical gardens, etc. |
| | Comprehensive park | The area is larger than 10 ha, rich in content, suitable for all kinds of outdoor interaction, with green space with complete facilities. |
| Park size | <2 ha park | Parks less than 2 hectares in size. |
| | 2–5 ha park | Parks of 2–5 hectares in size. |
| | 5–10 ha park | Parks of 5–10 hectares in size. |
| | >10 ha park | Parks of more than 10 hectares in size. |
| Park with or without water | Park with water | The park has water resources such as lakes, creeks, and rivers. |
| | Park without water | No water resources such as lakes, creeks, and rivers inside the park. |

**Table 3.** Description of landscape indices used in this study.

| Category | Metrics | Abbreviation | Description |
|---|---|---|---|
| Aggregation metric | Number of Patches | NP | Reflecting the spatial pattern of the landscape, the value is positively correlated with landscape fragmentation. |
| | Patch Density | PD | The density of corresponding patches within an analysis unit. |
| | Aggregation Index | AI | Degree of aggregation of the corresponding patches within an analysis unit. |
| | Contagion | CONTAG | Reflecting different patch types and clustering or extension trends in the landscape, small values indicate high landscape fragmentation. |
| | Interspersion and Juxtaposition Index | IJI | Reflecting the spatial pattern of the landscape, larger values indicate the proximity of patch types to each other and high dispersion. |
| | Patch Cohesion Index | COHESION | The measure of the physical connectedness of the focal land cover class. |
| | Splitting Index | SPLIT | SPLIT equals the total landscape area ($m^2$) squared divided by the sum of patch area ($m^2$) squared, summed across all patches of the corresponding patch type. |
| | Landscape Shape Index | LSI | Landscape shape index, landscape shape index of the landscape in the spatial unit. |

**Table 3.** *Cont.*

| Category | Metrics | Abbreviation | Description |
|---|---|---|---|
| Shape metric | Shape Index Distribution | SHAPE_MN | Average shape index of the corresponding patches within an analysis unit. |
| | Perimeter–Area Ratio Distribution | PARA_MN | Reflecting the complexity of landscape patch shapes and the extent to which land use is influenced by human activities. |
| | Mean Fractal Dimension Index | FRAC_MN | Average patch shape complexity measures approach 1 for simple shapes and 2 for complex shapes; it reflects shape complexity across various spatial scales (patch sizes). |
| Area and edge metric | Percentage of Landscape | PLAND | Landscape percentage of the corresponding patch. |
| | Largest Patch Index | LPI | The percentage of the landscape occupied by the largest patch. |
| | Mean Patch Area | AREA_MN | The average size of the patches. |
| Diversity metric | Shannon's Evenness Index | SHEI | Uniformity of distribution of landscape types. |
| | Shannon's Diversity Index | SHDI | Reflecting the abundance and complexity of landscape types. |

### 2.2.3. Quantification of Urban Parks' Carbon-saving Potential

This section aims to estimate the carbon emission reduction potential of urban green spaces by calculating the carbon emission reduction resulting from the alleviation of the urban heat island effect in city parks. Following the previous study [26,27], we calculate the carbon-saving intensity (CSI) and the carbon emission reduction efficiency (CSE):

$$CSI = k \cdot \rho \cdot a \cdot \int_0^H \sum_{i=0}^N \frac{1}{3}(S_i + S_{i+1} + \sqrt{S_i S_{i+1}}) \Delta T dh \qquad (2)$$

$$CSE = CSI \div S \qquad (3)$$

among them, k represents the specific heat at constant pressure, with a value of $1004.68 \text{ J kg}^{-1} {}^\circ\text{C}^{-1}$, $\rho$ represents the air density ($1.2923 \text{ kg m}^{-3}$), and a is the conversion coefficient of energy consumption into carbon emissions from coal-fired power generation (841 g/3.6 MJ). h represents the vertical influence range (H = 70 m). $\Delta T$ is the temperature difference between adjacent buffer zones, $S_i$ and $S_{i+1}$ are the base areas, and S is the park area.

### 2.2.4. Correlation between Urban Parks' Carbon-Saving Potential and Landscape Metrics

This study applied various statistical analysis methods to explore the relationship between carbon-saving potential and landscape pattern index in urban parks. These methods include the Shapiro–Wilk test, One-way analysis of variance (ANOVA), Pearson correlation analysis, and stepwise multiple linear regression. Firstly, the Shapiro–Wilk test is used to ensure that the sampled data set conforms to a normal distribution. Secondly, ANOVAs are used to analyze whether there are significant differences in the carbon-saving potential of urban parks under different park classifications. Thirdly, Pearson correlation analysis is used to determine whether there is a statistically significant correlation between various landscape indices and carbon-saving intensity, as well as if the correlation is obvious. Finally, stepwise multiple linear regression is used to establish the relationship between carbon-saving intensity and landscape patterns, the best-fitting model is selected, and the significance of the coefficients is determined based on regression statistics ($R^2$, *p*-value) [10]. In conclusion, these methods provide a comprehensive way to analyze factors

that influence carbon-saving intensity in urban parks and can be used to propose strategies for achieving carbon neutralization.

## 3. Results

### 3.1. Spatial Heterogeneity of Carbon-Saving Potential in Different Urban Parks

For CSI, Shangqiu City Park green spaces (538 ha) can reduce a total of 300.57 t $CO_2$ emissions. Specifically, in the parks of Shangqiu, the CSI distribution range is 0.04~18.93 t, and on average, each park can save 2.55 ± 0.31 t $CO_2$ emissions for the city because of its contribution to the mitigation of surface heat effects, of which the canal ribbon park has the highest CSI, reaching 18.93 t $CO_2$. Among the parks divided according to different classification criteria, special parks (3.42 t), parks larger than 10 ha (6.94 t), and parks with water bodies (4.71 t) had the largest CSI average in their respective classifications. From the efficiency perspective, the CSE distribution in Shangqiu City was 16.34~0.04 t $CO_2$ ha$^{-1}$, and the average CSE was 1.79 ± 0.29 t $CO_2$ ha$^{-1}$. Among them, small parks, parks less than 2 ha, and parks without water showed a higher trend of average CSE. In other words, the CSE was larger for smaller parks, with the highest CSE (16.34 t $CO_2$ ha$^{-1}$) found in Chinese Phalarope Square (0.10 ha). Figure 3 shows the results of the analysis of the difference between CSI and CSE in different classifications of parks, and there are certain differences in the performance of CSI in different types of parks, but the CSE of each park does not show significant differences, and this difference is not reflected in all park classifications. According to the classification of different areas, the CSI of parks less than 2 ha in size is significantly lower than that of parks with an area of 5–10 ha and parks larger than 10 ha, the CSI of parks with an area of 2–5 ha is significantly lower than that of parks with an area greater than 5 ha, and the CSI of parks with an area greater than 10 ha is significantly larger. Among the parks classified according to whether the park contains water bodies, the CSI of parks with water is significantly higher than that of parks without water.

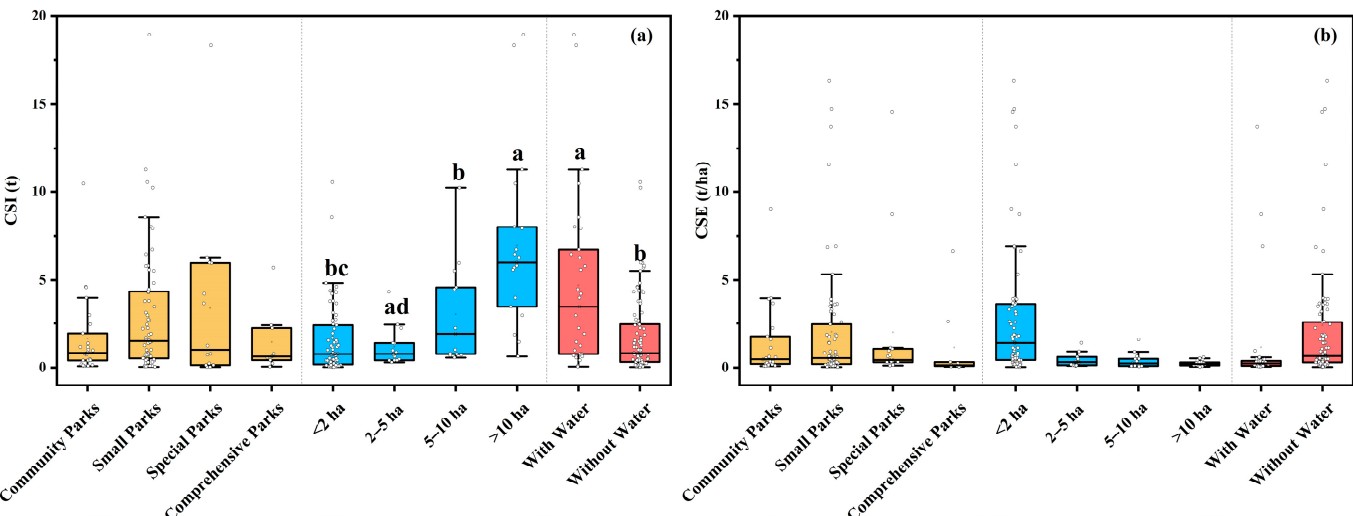

**Figure 3.** Park category differences between CSI (**a**) and CSE (**b**). a–d represent significant differences determined by Fisher's least significant difference (LSD) tests (*p* < 0.05) on different seasons for different park types.

### 3.2. Spatial Changes in Landscape Metrics

In the study area, the park area was distributed in 0.03~357.00 ha, and the proportion of various landscapes in the park was different, from large to small; they were woodland (59.42%), grassland (23.09%), impervious surface (22.20%), and water body (12.34%). We selected 15 landscape pattern indexes to explore the differences in surface landscape patterns of various parks, and on the whole, Shangqiu City Park has a high degree of fragmentation, a high degree of aggregation, and landscape diversity, but the shape of various landscape

patches is more complex. In different types of parks, there are certain differences in landscape indicators (Figure 4). The analysis of the differences in the landscape pattern indices of different parks shows that the landscape indices do not show significant differences when the parks are divided according to the park category, and among the parks with different areas, the AI, SHDI, and COHESION of parks with an area of less than 2 ha were significantly larger than those with an area greater than 2 ha, and COHESION was the opposite. Parks with an area greater than 5 ha SHAPE_MN were significantly larger than parks with an area of less than 2 ha, LSI and NP were significantly larger than parks with an area of less than 5 ha in parks with an area of 5–10 ha, and LSI, NP, and PLAND_Imper were significantly smaller in parks with an area of less than 2 ha. LSI, NP, SPLIT, and AREA_MN increased significantly with the increase of park area when the park area was less than 10 ha, and on the contrary, PD decreased significantly with the increase of area when the area was greater than 2 ha. For parks with and without water, the values of AI, LPI, COHESION, PD, and PLAND_Forest in parks with water were significantly greater than those in parks without water, while SHDI, SPLIT, AREA_MN, LSI, and NP were the opposite, and their values in parks without water were significantly larger.

### 3.3. The Relationship between the Carbon-Saving Potential and Landscape Driving Factors in Different Urban Parks

There were similarities and differences in the correlation results between various park landscape indexes and CSI (Figure 5). On the whole, CSI was significantly positively correlated with SPLIT, IJI, AREA_MN, LSI, NP, and PLAND_Water, and significantly negatively correlated with LPI and PLAND_Imper. Among the different types of parks, the CSI of AREA_MN and comprehensive parks (−0.952) and community parks (−0.869) showed a strong and significant positive correlation, while PLAND_imper was the opposite. The CSI of the garden was positively correlated with NP, IJI, and SPLIT, but negatively correlated with LPI. The CSI of special parks showed a significantly strong positive correlation with AREA_MN (0.952), NP (0.92), and PLAND_water (0.905), while it showed a significantly strong negative correlation with FRAC_MN (−0.989), and the CSI of comprehensive parks showed a correlation with the most landscape index, among which there was a strong positive correlation with AREA_MN (0.952) and CONTAG (0.856) and a strong positive correlation with PLAND_Imper (−0.957); PLAND_Grass (−0.818) showed a strong negative correlation. Among parks of different sizes, the number of landscape indices associated with CSI was slightly smaller than in the other two categories, but the correlation between them was strongest. The CSI of 2–5 ha parks was significantly positively correlated with SHEI (0.995), SHDI (0.995), and PLAND_Grass (0.998), the CSI and AREA_MN of parks with an area greater than 2 ha showed a significant positive correlation, and the CSI of parks with an area greater than 10 ha showed a different correlation from other parks; the CSI was significantly positively correlated with NP (0.717) and PLAND_Water (0.813), and it was significantly negatively correlated with PLAND_imper (−0.591). In the classification according to whether the park has a water body, there are more landscape pattern indices related to CSI, and the correlation between parks with water and parks without water and landscape pattern indexes is similar, and there are significant positive correlations with SPLIT, AREA_MN, LSI, and NP, among which the positive correlation between CSI and NP in parks with water is the strongest (0.733), while the negative correlation between parks without water and LPI is the strongest (−0.385). The difference is that the significant correlation between IJI (0.37), PLAND_Water (0.644), and PLAND_Imper (−0.348) and parks with water is not reflected in parks without water.

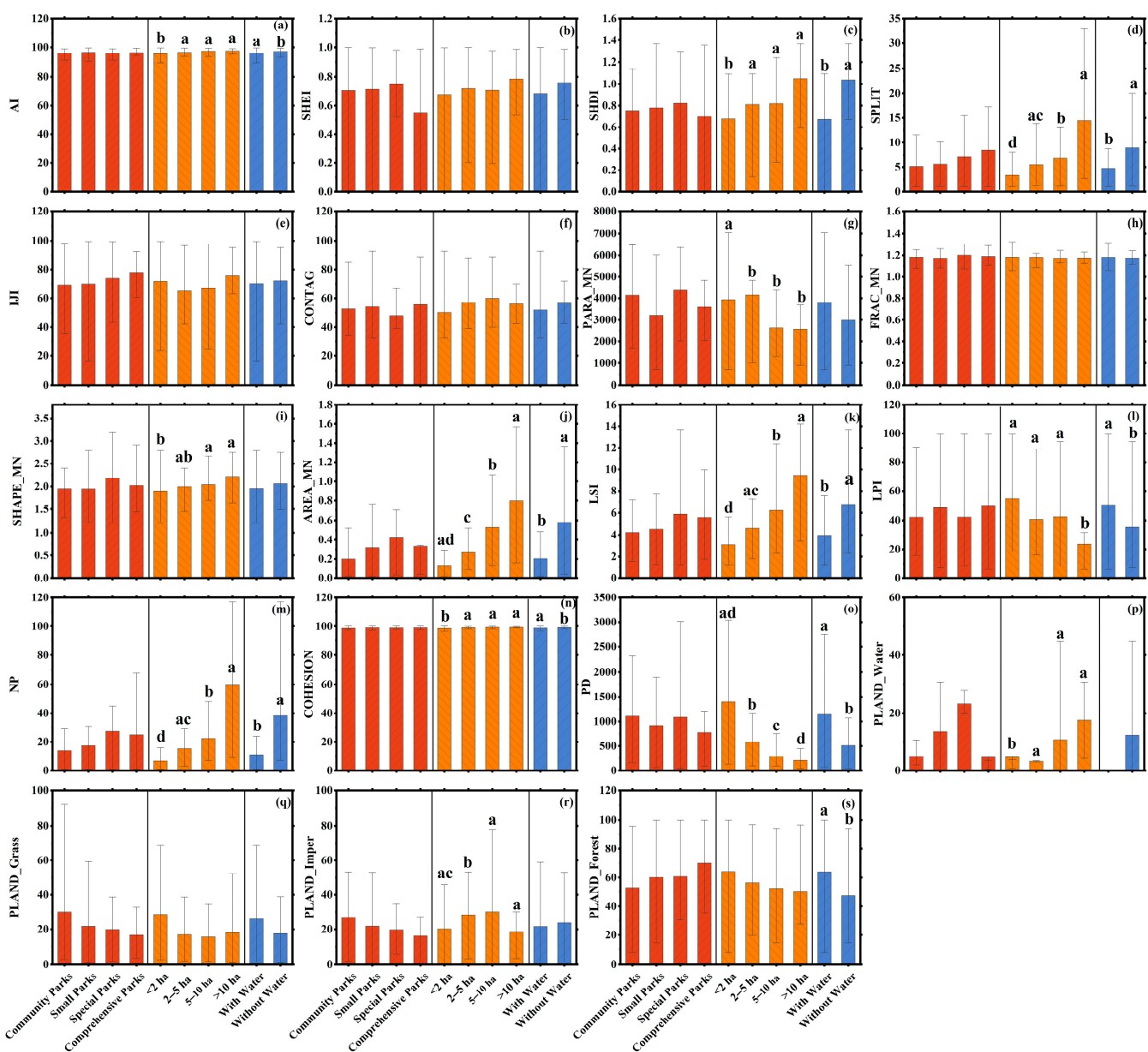

**Figure 4.** Park category differences of landscape patterns (**a–s**). a–d represent significant differences determined by Fisher's least significant difference (LSD) tests (*p* < 0.05) on different seasons for different park types.

Figure 6 shows the correlation between CSE and landscape pattern index of various parks in Shangqiu City, and the results show that the correlation between CSE and landscape pattern index is similar to the CSI trend overall, both in parks classified by category, the most relevant landscape pattern index, and by parks classified according to water bodies, and the correlation between CSE and landscape pattern in parks classified by size is weak. The difference is that the landscape pattern index related to CSE of various parks is completely different from CSI, and AI, COHESION, and park CSE are significantly negatively correlated, while PARA_MN and PD are significantly positively correlated with park CSE. Among different types of parks, the landscape pattern indexes related to CSE of community parks were SHEI (0.809), SHDI (0.809), PARA_MN (0.831), and PLAND_Grass (0.945), respectively, and all showed positive correlations. In contrast, CSE in comprehensive parks was only significantly negatively correlated with PLAND_Grass (−0.836). In small parks and special parks, CSE had a significant negative correlation with AI and COHESION and a positive correlation with PARA_MN and PD, with the latter

correlation being stronger in special parks. In addition, SHAPE_MN (−0.458), AREA_MN (−0.466), and small park CSE were significantly negatively correlated, while special park CSE and LSI were negatively correlated and positively correlated with PLAND_Imper (0.986). Among the parks with different areas, park CSEs less than 2 ha were positively correlated with NP (0.85), and park CSEs with 5–10 ha also showed a positive correlation with AREA_MN (0.734). Among parks with water and parks without water, the correlation between landscape pattern index and CSE was more consistent and significantly negatively correlated with AI and COHESION; this correlation was stronger in parks with water, the significant positive correlation with PD was stronger in parks without water, and the difference was reflected in the significant positive correlation between CSE and PARA_MN (0.406) in parks with water, while CSE in parks without water was positively correlated with PLAND_Grass (0.292).

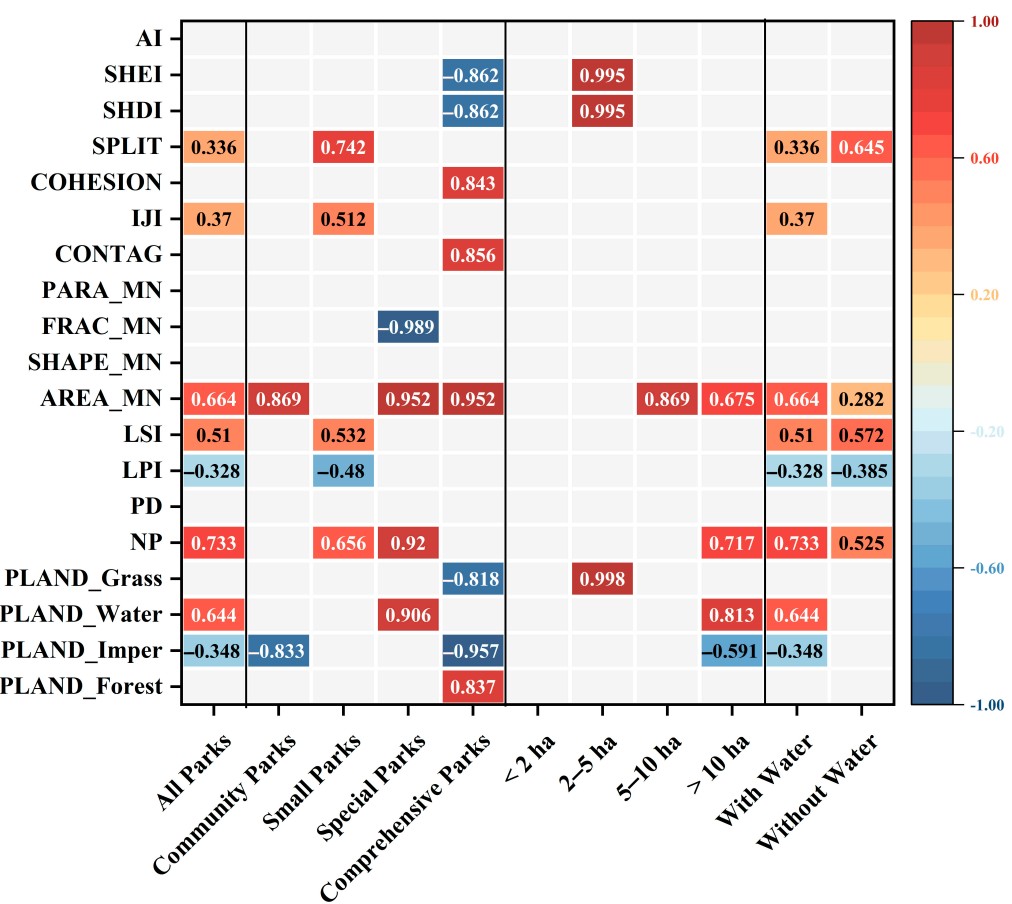

**Figure 5.** Heatmaps of correlation coefficients between landscape patterns and CSI in different types of parks.

*3.4. Identify the Landscape-Driving Factors*

For all the landscape pattern variables examined in our study, the direction and magnitude of their impact on park CSI and CSE were generally different (Figures 7 and 8). Moreover, the specific influencing factors are different from the direction and degree of their influence on CSI and CSE to different park categories, and the overall changes in CSI and CSE in different parks have a high degree of explanation. The fragmentation index has a more common effect on the changes in CSI and CSE in various parks, while the explanatory degree of the diversity index on CSI and CSE changes is significantly higher than that of other factors. The overall interpretation rate of CSI of various landscape pattern indexes can reach 83%, of which NP contributed the largest 52% of the interpretation rate, followed by AREA_MN (17.39%), PD (9.41), and FRAC_MN (4.2%). Among the different types of parks, the regression model of CSI for comprehensive parks was the best, with a

goodness of fit of 99%, and the fitting degree of the model for special parks was also high, at 98.03%. Among them, PLAND_Imper contributed the highest explanation rate (88.7%) for the CSI of the comprehensive park, PARA_MN contributed 10.3% of the explanation rate, the largest contributor in the regression model of the special park was FRAC_MN (94.7%), and SPLIT also explained a small part of the CSI change. The CSI model fit was slightly weak, but the influencing factors were relatively balanced: SPLIT contributed 51.8% of the explanation rate, and PLAND_Forest contributed 11.8%. In parks with different areas, except for parks with an area of less than 2 ha, the park model with a smaller area has a higher fit and clearer influencing factors. The park regression model of 2–5 ha had the highest goodness of fit (98.6%), and the vast majority of explanatory degrees were contributed by PLAND_Grass (97.8%), while SHDI explained the model only 0.8%. For 5–10 ha parks, two variables, AREA_MN and LSI, were introduced into the regression model, with explanatory rates of 75.5% and 22.2%, respectively. Among the parks with an area greater than 10 ha, the interpretation of CSI was PLAND_Water (63.7%), AREA_MN (13.82%), and NP (5.36%), respectively. Compared with the above classification, the CSI driving factors of the parks with water and parks without water were scattered, and the four influencing factors, FRAC_MN (4.16%), AREA_MN (17.39%), PD (9.41%), and NP (52%), were introduced into the parks with water's regression model; the overall model goodness of fit reached 83%. The goodness of fit of the CSI regression model for parks without water was 50.5%, of which SPLIT contributed 40.4% of the explanatory degrees.

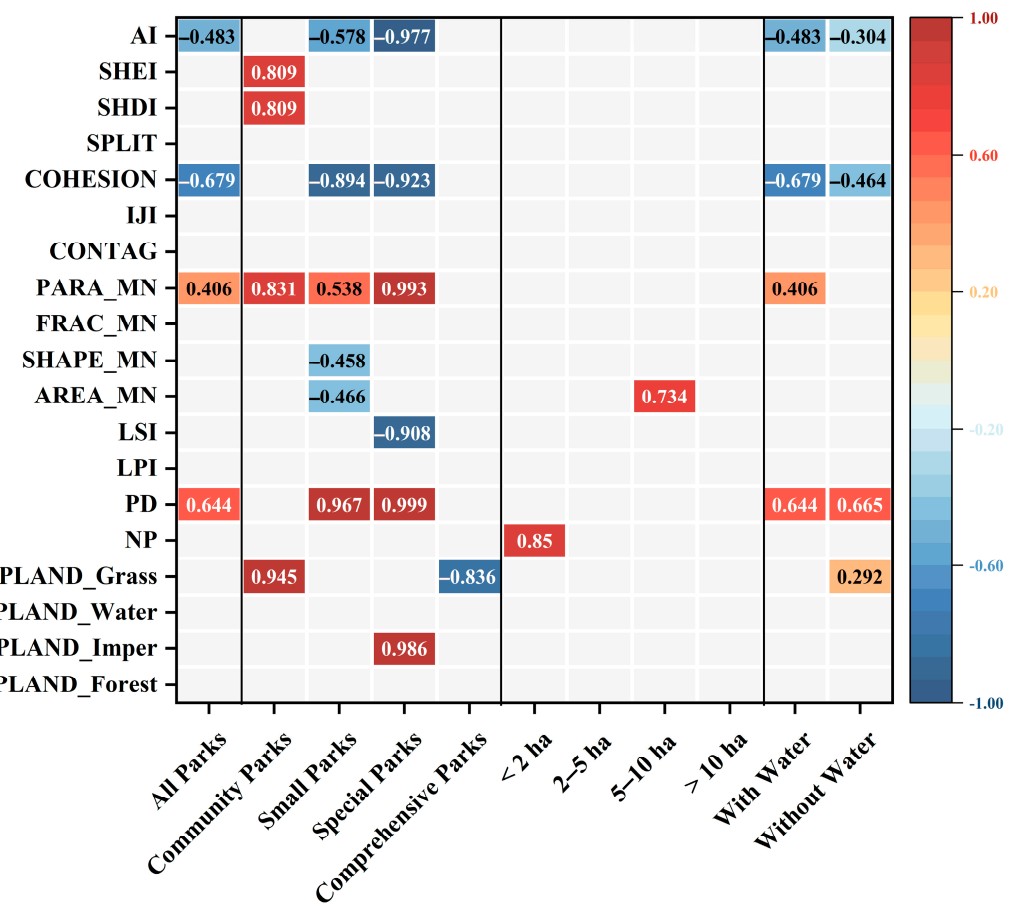

**Figure 6.** Heatmaps of correlation coefficients between landscape patterns and CSE in different types of parks.

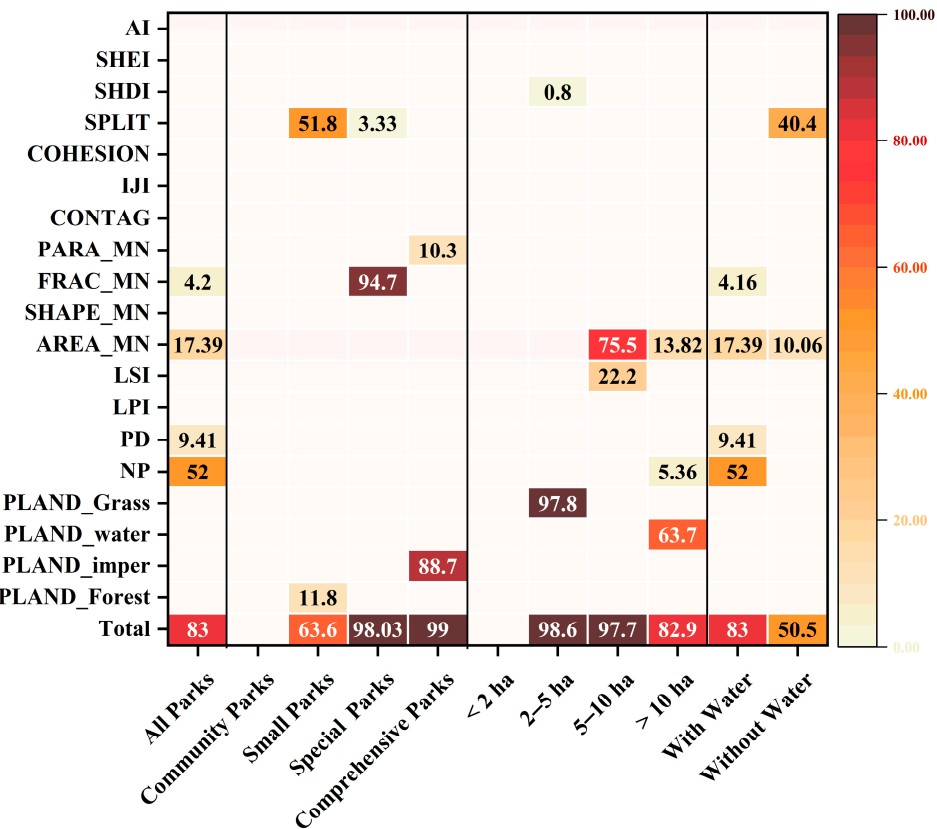

**Figure 7.** Stepwise regression diagram of landscape pattern and CSI in different parks.

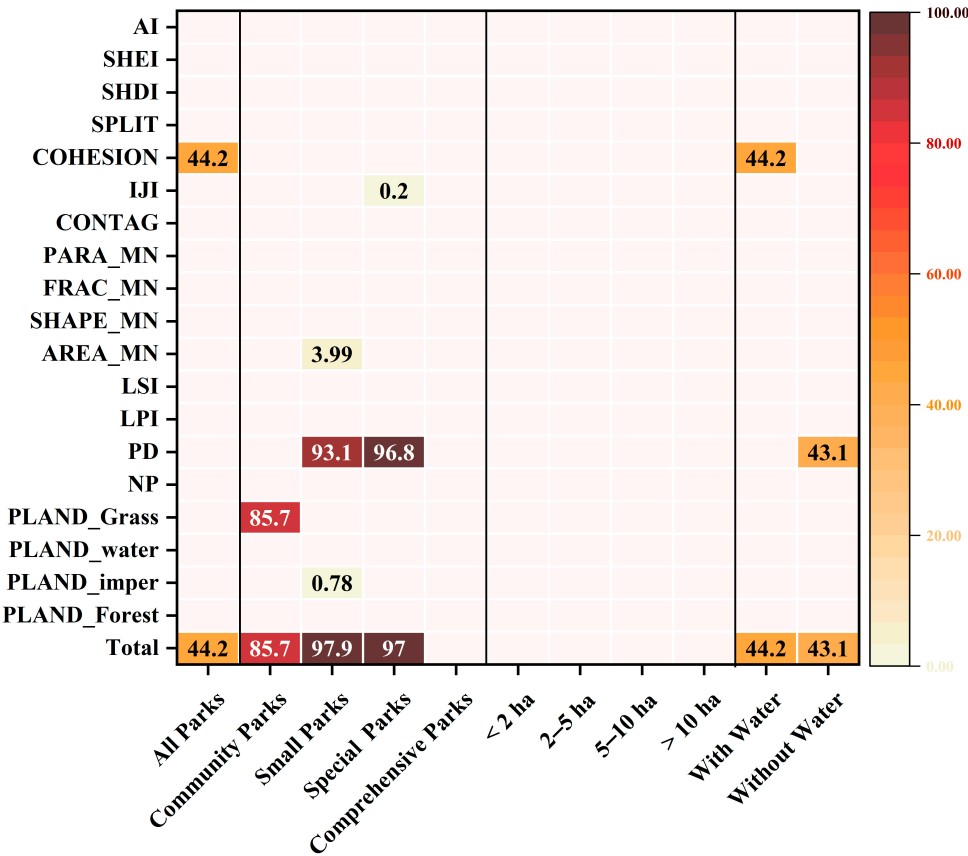

**Figure 8.** Stepwise regression diagram of landscape pattern and CSE in different parks.

The results of CSE stepwise regression in various parks (Figure 8) showed that compared with the CSI regression model, the CSE model introduced fewer variables, and the overall explanatory rate was not high. All parks were included in the model, and the model introduced COHESION as the only explanatory variable, with an explanatory rate of 44.2%. Among the different types of parks, community parks also introduced PLAND_Grass as the only explanatory variable, but the explanatory degree reached 85.7%, and PD in small parks and special parks were the most explanatory landscape pattern indicators, with 93.1% and 96.8% explanatory degrees, respectively. The goodness of fit of the small park CSE regression model was 97.7%, and in addition to PD, AREA_MN and PLAND_imper also contributed 3.99% and 0.78% of the explanatory degrees, respectively. IJI was also introduced into the special park CSE model to provide an explanatory rate of 0.2%. Classified by area, none of the factors were introduced into the regression model due to confounding factors. COHESION and PD were introduced as the only explanatory variables in the regression model of the park with water and park without water, and the goodness of fit was 44.2% and 43.1%, respectively.

## 4. Discussion

### 4.1. The Carbon-Saving Potential of Urban Parks

The results showed that the average CSE of Shangqiu City Park was $1.79 \pm 0.29$ t $CO_2$ ha$^{-1}$, which was higher than the previous CSE study of the Yangtze River Economic Belt City Park in China ($1.08 \pm 0.03$ t $CO_2$ ha$^{-1}$). First, studying the size difference of the region may be one of the reasons for the difference in CSE. The study area is a city-level city in Henan Province, with a total of 118 parks. The previous study area included 1510 parks in 26 cities, spanning 11 provinces in China. A higher number of parks studied directly means greater differences in size between parks, many types of parks, and more complex basic information about parks. So, it is understandable that the average CSE for 1510 parks is lower than the average CSE for 118 parks. On the other hand, the size of the study area will cause differences in the urban LST results, and may also indirectly affect the calculation results of CSI and CSE. For example, a study of Bangkok, Jakarta, and Manila showed that LST studies presented different results at different spatial resolutions, and proposed that 210 m × 210 m is an optimal characteristic area or land climate footprint that can be used for examining any meteorological, climatic, or environmental issues in urban areas or for landscape and urban planning [35]. The results for the 960 m scale and the 240 m scale are completely different [32]. It is worth noting that the previous research results also show that the average CSE of small urban parks is generally high, which is partly consistent with the results of this study.

Second, the climate type of the study area is also responsible for the difference in CSE. Previous results on the Yangtze River Economic Belt showed that the average CSE of parks with humid subtropical climates was higher than that of parks located in humid subtropical monsoon climates. The climate of Shangqiu City is a temperate monsoon climate, and the difference in climatic characteristics is more obvious. Different climatic characteristics lead to different LST research results and indirectly lead to different CSEs [10,36].

Third, studying differences in regional economic development, population density, and urbanization levels will also lead to differences in the average CSE of parks. Several studies have pointed out that economic development, population growth, and urbanization are important causes of rising surface temperature [37]. Higher levels of urbanization mean more natural features, such as vegetation and water bodies, that have been replaced by impervious materials and buildings [38]. Economic activity and population agglomeration consume large amounts of fossil fuel resources and exacerbate the urban thermal environment [39]. Compared to Shangqiu City, the Yangtze River Economic Belt, the former study area, has the most prosperous and dense urbanization performance in China, which also means a denser population [40]. In contrast, Shangqiu City, Henan Province, located in central China, is lower than the Yangtze River Economic Belt in terms of economy, population density, and urbanization level. This directly leads to the different urban heat

island effects between the two, and also indirectly leads to the difference in the average CSE of the park.

Fourth, Shangqiu City's emphasis on park construction has also greatly increased the average CSE of the park. Shangqiu City was awarded the honorary title of "National Garden City" as early as 2010. Shangqiu City uses the "14th Five-Year Plan for Urban Renewal and Urban and Rural Habitat Environment Construction of Shangqiu City" as its program, intending to build a green city and ecological city by continuously strengthening the urban landscaping construction in Shangqiu City, continuing to pay attention to the incremental quality improvement in park green spaces, and improving the carbon emission reduction capacity of Shangqiu City Park.

In addition, this study shows that the annual CSE of the park green spaces in Shangqiu City is 154.8 t $CO_2$ ha$^{-1}$ year$^{-1}$. Another result of our ongoing study shows that the annual average carbon serum in Zhengzhou Parkland is 19.07 t $CO_2$ ha$^{-1}$ year$^{-1}$. Another study based on the rates of major Chinese cities showed that the average annual carbon sequestration of green infrastructure in Zhengzhou was 10.52 t $CO_2$ ha$^{-1}$ year$^{-1}$ [41]. This study showed that CSE in Shangqiu was 8 and 14 times higher than the two studies on green space carbon sequestration in Zhengzhou, respectively. This strongly indicates that the cooling effect of the park's green space leads to significant carbon reduction.

*4.2. Effects of Landscape Patterns on the Carbon-Saving Potential*

In general, the impact of the park's landscape pattern on the park's carbon reduction potential is significant. The park's carbon reduction capacity is equivalent to the park's cooling and energy-saving effects. This result is, therefore, consistent with previous studies, confirming the cooling effect of landscape patterns on parks [10,42–45]. Specifically, the main drivers of CSI in Shangqiu City Park are NP, AREA_MN, and PD. In other words, the number of park patches and the degree of landscape fragmentation (characterized by AREA_MN and PD) significantly affected the park CSI. For the number of patches (NP), it can be interpreted that smaller, more numerous plaques in the park can play a role in cooling energy saving and carbon reduction. A study of 197 water bodies in Beijing showed that tiny lakes and ponds play an important role in cooling. Therefore, it is suggested that decomposing parks with large water areas into smaller ones can improve the cooling effect of parks, and the same applies to park carbon reduction [46]. The impact on landscape fragmentation depends on the type of land use. For example, the AREA_MN of impervious surface panel blocks has an increasing effect on LST, while plant plaques have the opposite effect. Today's increased urbanization has led to the fragmentation of impervious surfaces and patches of green space, resulting in the fragmentation of the urban landscape pattern. By reducing the fragmentation of the urban landscape, urban LST can be lowered, and the carbon reduction potential of parks can be increased [45].

In addition, the impact of the PLAND index on the CSI and CSE of Shangqiu City parks was shown in different categories of parks, including PLAND_Grass, PLAND_Water, PLAND_Imper, and PLAND_Forest. This result is supported by many LST studies [32,47,48]. Even if the study area is different from the meteorological environment, the enhancement effect of PLAND_Imper on park LST and the cooling effect of PLAND_Grass, LAND_Forest, and PLAND_Water on LST due to high reflectivity, transpiration, and providing shade to reduce cooling and its specific heat capacity have been widely demonstrated [18,49–51]. A park's carbon reduction potential, which is closely related to the park's cooling and energy-saving effects, will also be affected by the PLAND index.

But, contrary to Hao Hou's index, where Shape_MN is the best-performing cooling effect [43], Shape_MN does not affect Shangqiu City's CSI. This inconsistency is normal in LST studies. Different study areas and different climatic conditions significantly affect the performance of landscape indicators [32,52,53]. For example, Shangqiu City has a typical temperate continental monsoon climate, but Hangzhou City has a subtropical monsoon climate, and there are significant climatic differences between the two. Therefore, when exploring the relationship between landscape patterns and the effects of park cooling energy

saving and carbon emission reduction, it is necessary to refer to the natural environmental factors of the study area [52–54]. Second, the land use information in this study is two-dimensional and does not involve factors such as urban elevation or building height. These may also be the reasons for the different results. In more detail, [55] results showed that densely built-up areas had lower LST acting as cool islands, which also implies a higher carbon-saving potential. The reason is that buildings reduce surface albedo horizontally and alter wind turbulence vertically, thereby reducing the diffusion of heat and pollutants [56].

*4.3. Implications for Urban Planning and Management*

This study explores the impact of landscape patterns on CSI and CSE of different types of urban park green spaces, the specific relationship, and its potential value in reducing carbon emissions from the perspective of carbon emission reduction caused by the cooling effect of green space. Rapid urbanization has caused an imbalance in land resource distribution, leading to the need for rational allocation of the internal landscape pattern of parks. This is important to maximize the cooling effect of urban green spaces and achieve the strongest carbon emission reduction efficiency in the context of carbon neutrality in China. This study focuses on several parks and green spaces in the study area to understand the impact of the internal landscape pattern on the efficiency and intensity of urban carbon emission reduction at the regional scale. This will provide guidance and suggestions for upgrading and renovating urban parks. Previous research shows that landscape pattern change greatly affects the surface thermal environment of urban parks and reduces carbon emissions. Urban planners can explore the strongest mode of urban park green spaces to reduce carbon emissions by updating and transforming the urban park landscape pattern through design.

The results showed that the CSI and CSE of the park differed with the change in landscape pattern, and this difference was manifested in different types of parks. Overall, FRAC_MN, PLAND_Forest, PLAND_Grass, and NP were all significant influencing factors with positive effects on CSI and CSE. This indicates that parks with a larger proportion of trees and ground cover plants, as well as more complex patch shapes, have a better potential for carbon saving. These findings are consistent with previous research in the field of the thermal environment [57–59]. The proportion of water in parks has a significant positive impact on CSI, as water bodies have a high specific heat capacity. Small lakes or ponds, in particular, play an important role in carbon emission reduction [60]. Parks with water, special parks, and parks larger than 10 ha show significant carbon reduction effects. Therefore, water design and transformation should be the focus of upgrading and renovation in the aforementioned types of parks. Our research results indicate that changes in various landscape patterns in different types of parks have varying degrees of impact on carbon emission reduction. This paper proposes targeted park improvement and updated suggestions for park categories with relatively low carbon emission reduction in the study area. These include reorganizing landscapes and changing the area and location of different landscape types. Suggestions for transforming different types of park green spaces with weaker carbon emission reduction capacity were also proposed to adjust the density of patches and the shape of parks, to accurately promote the carbon emission reduction in park green spaces in Shangqiu.

The diversity index has a significant positive impact on the CSE of the integrated park, and the increase in PLAND_Imper can bring about a significant decrease in CSI. Therefore, when transforming comprehensive parks, the focus should be on reducing the proportion of impervious surface area in the park and increasing the diversity of landscape patches. This can include introducing a variety of plant communities into a single green space and adding small water bodies, such as fountains and pools, to parks without water. The CSI of LSI of a small park has a significant negative impact, and the complexity of the park boundary provides an opportunity for energy exchange between the park and the surrounding area, thereby increasing the cooling effect of the park to a certain extent [26], so the design and transformation of the park should focus on the change in the shape of the

park boundary. The increase of AREA_MN and PLAND_Grass can significantly improve the carbon emission reduction capacity of parks and community parks with an area of 2–5 ha [61], so attention should be paid to improving the cover of surface grassland in the renovation of such parks, to enhance the carbon storage of vegetation and thus improve the carbon emission reduction capacity of parks. For parks with an area of 2–5 ha, it is also possible to improve the carbon-saving capacity of the park by increasing the type of landscape patch. In the park without water, PD, SPLIT, and AREA_MN all showed a significant positive relationship with carbon emission reduction capacity, which indicates that the degree of plate fragmentation has the most significant impact on the sewage park, so attention should be paid to dividing the internal patches of the park to make them as dispersed as possible to achieve greater carbon emission reduction efficiency.

*4.4. Limitations and Future Research Directions*

In this paper, 19 landscape pattern indicators were selected to comprehensively describe the morphology, patch characteristics, fragmentation degree, and aggregation degree of the park, and the selected variables explained the changes in CSI (83%) and CSE (44.2%). However, our study has some limitations in some aspects, and the influence mechanism of landscape patterns on CSI and CSE of urban parks and green spaces needs further study. First of all, it should be recognized that this study is based on a 2D plane, and the impact of landscape changes in the vertical 3D range, including green amount, water depth, and other factors on CSI and CSE, needs to be refined and improved in future studies. The specific configuration of plant communities in the park, the physiological and ecological indicators of vegetation, the vertical and planar structure of surrounding features, and other factors that have a potential impact on the cooling effect of urban park green spaces are also included in the research of carbon emission reduction and park enhancement strategy of urban parks. The impact of influencing factors on carbon emission reduction from various perspectives should be comprehensively considered.

Secondly, the research on the carbon emission reduction effect of urban parks in this paper is based on surface temperature. Due to the difficulty in acquiring high-quality continuous meteorological data, this paper relied only on remote sensing images of cloudless and sunny summer days in 2020–2021 to measure the carbon emission reduction in urban park green spaces. Actual results may be biased due to this limitation. Future research should explore higher precision surface temperature data combined with field survey data to supplement remote sensing images. It should also introduce multi-source data from seasonal changes in surface temperature, daily dynamic changes, interannual changes, urban and rural changes, and other multi-temporal and spatial perspectives to explore the changes in land surface temperature and its impact on carbon emission reduction efficiency in a comprehensive manner. This approach will clarify the specific impact of landscape pattern index and urban park CSI and CSE and will direct attention to the practical significance and practical effect of research. It will also promote the combination of research results and urban construction strategies and conduct in-depth research on the carbon emission reduction effect of urban green spaces in practice.

## 5. Conclusions

Currently, there is extensive research on mitigating the surface heat island effect in urban parks in the context of carbon neutrality. However, it is of significant importance to quantify the carbon emission reduction resulting from the mitigation of urban park thermal effects, as it contributes to achieving urban carbon neutrality goals. Additionally, it is crucial to quantitatively analyze the influence of urban park landscape patterns on carbon emission reduction intensity and efficiency. In this study, we estimated the carbon emission reduction intensity and efficiency of 118 urban parks located in Shangqiu City, Henan Province. The average CSI was found to be $2.55 \pm 0.31$ t $CO_2$, and the CSE was $1.79 \pm 0.29$ t $CO_2$ per ha across all studied parks. Consequently, a total carbon emission reduction of 300.57 t $CO_2$ was achieved. Parks larger than 10 ha and parks with water

features exhibited higher carbon emission reduction, indicating that the landscape layout of these parks is more conducive to carbon reduction compared to other park types. Among all park categories, the proportion of trees, herbs, and water bodies significantly influenced carbon conservation. Furthermore, the concentration of landscape patches, including cohesion, split, and aggregation index (AI), played a crucial role in the CSI and CSE of urban parks. Increased fragmentation also led to a stronger carbon emission reduction effect. Based on these findings, we propose a series of strategic suggestions for the renovation and improvement of different park types, aiming to enhance the carbon emission reduction intensity of urban park green spaces through landscape pattern transformations at the regional scale. These suggestions provide theoretical support and practical guidance for urban planning, renovation, and renewal efforts, contributing to carbon emission reduction, mitigation of the urban thermal environment, and enhancement of ecological benefits in the Central Plains region. Ultimately, this research promotes the acceleration of the carbon neutrality process in the Central Plains.

**Author Contributions:** Conceptualization, J.G. and S.G.; methodology, J.G., H.H. and S.G.; software, H.H. and S.G.; validation, J.G. and H.H.; formal analysis, S.G. and H.H.; investigation, S.G. and H.H.; resources, J.G. and S.G.; data curation, J.G., H.H. and S.G.; writing—original draft preparation, J.G., H.H. and S.G.; writing—review and editing, J.G., H.H. and S.G.; visualization, J.G. and S.G.; supervision, J.G. and S.G.; project administration, J.G. and H.H.; funding acquisition, J.G. and S.G. All authors have read and agreed to the published version of the manuscript.

**Funding:** This study was supported by the Social Science Fund of Tianjin of China grant (TJYY19-013), the Key Technology R&D Program of Henan Province (232102320190), the Center for Blockchains and Electronic Markets funded by the Carlsberg Foundation under grant no. CF18-1112, and the Special Fund for Young Talents in Henan Agricultural University under grant no.30500930.

**Data Availability Statement:** The data presented in this study are available on request from the first author.

**Conflicts of Interest:** The authors declare no conflict of interest.

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
