# Peer review of "Carbon-Saving Potential of Urban Parks in the Central Plains City: A High Spatial Resolution Study Using a Forest City, Shangqiu, China, as a Lens"

_land, doi:10.3390/land12071383_

Round 1

Reviewer 1 Report

Dear authors, thank you for your contribution. 

"Carbon saving potential of urban parks in the Central Plains 1 City: A High Spatial Resolution Study Using a Forest City, 2 Shangqiu, China as a Lens" is a study on the carbon-saving potential of urban parks in Shangqiu City that uses high-resolution remote sensing images.

The objectives of the study are clearly stated in the document. The general aim is to provide scientific research input to guide modern urban carbon neutrality strategies. While, on a specific level, the research questions posed are: What are the magnitude and spatial distribution of carbon-saving potential in Shangqiu's urban parks? What are the landscape factors driving spatial heterogeneity in the carbon-saving potential across different urban parks in Shangqiu?

The results are well presented and are complemented by interesting comments on the case study.

However, the contribution presents some critical issues and can be improved.

1. The methodology used and the steps followed to obtain the results are not expressed in a clear and well-structured way. It might be useful for the reader to have a summary of the steps followed in the research (perhaps by using a bulleted list) to identify the stages of the work better and link them to the paragraphs (and subsections) of the paper. In this regard, it is advisable to add a sub-section before presenting the case study. This would also improve the reading of the document and allow any specific points of interest to be identified more quickly.

2. The literature review on the benefits of green areas in urban parks in mitigating the heat island effect is not exhaustive. 

3. The motivation for choosing Shangqiu as the study area of the research paper is unclear and not exhaustive.

4. Sub-section 2.3 does not explain how the "landscape indices" used in the work were selected. In addition, there is no bibliographical reference in this regard. 

5. In subsection 2.3 the presentation of the "landscape indices" could be improved and made more readable (e.g. by formatting the text differently or inserting a table), given their importance in the work.

6. The methods mentioned in section 2.5 and used in your work should first be presented within the methodological framework and be correlated by bibliographical references. In addition, they should be better described.

7. The quality of Figures 2 and 3 should be improved.

I wish you all the best.

Reviewer 2 Report

The cooling effect of urban green spaces may be one measure to reduce the need for air conditioning. However, the study presented here does not provide sufficient insights into the methodology to in fact demonstrate a) how parks influence the air temperature and b) how this may in turn have implications for cooling demands. From the existing literature, it is clear that the relation between land surface temperature and air temperature is complex and varies with urban morphology, building materials, vegetation, and meteorolgical conditions including wind and atmospheric stability. It is hence not sufficient to simply introduce equations 2 and 3 to then state that the CSI and CSE were being "measured" (e.g. line 484). There is a large number of underlying assumptions here that are not at all elaborated in the current state of the manuscript. Some not even discussed at all, such as the link between LST and air temperature, the meaning of the "vertical influence range" or the definition of "adjacent buffer zones". Given these drawbacks of the study design, all conclusions and discussions can not be interpreted appropriately.

minor comments:

Line 56-67: what are these definitions of Shangqiu city?

Line 110 and Lines 123-128: Not at all clear what the definitions of the park classes mean and how they should be interpreted.

Line 174-175: Not clear how these conclusions are drawn. How can you detect the CSE of an amusement park? Why would that be higher? Should you not expect high energy use in an amusement park? Again not clear how the appropriate emissivity correction is applied to account for surface structure of a amusement park.

Line 184: You state water-bearing parks have greater CSI, by which you really mean they are cooler? How did you account for emissivity correction of the water body in the LST product?

Line 202: Of course parks with a greater area are larger. This is not a scientific finding.

Line 342: given the definition of CSE in equation 2, obviously CSE is inversely proportional to park size. This is by definition and should not be stated as a scientific conclusion.

Line 347: the context of climate conditions and their relation to LST and also carbon emissions needs to be discussed in more detail. Probably building density, building age, etc also plays a very important role.

Please check for logical argument in the text and also repetition.  e.g. line 174-175: "... parks less tahn 2ha showed larger CSE..., while smaller parks showed larger CSE"

Reviewer 3 Report

The reviewed article is very interesting and touches on a very important problem, which is the need to improve microclimate conditions in an urbanized area. An important role in this process is played by green areas, especially parks, which were the main object of research, the results of which are presented in the article. The research methods were selected correctly and exhaustively described in the methodological part. The results were presented in a very accurate way (maybe even too much). In the discussion, the authors well explained the differences in the CSE scores obtained by different authors. Significantly, they also outlined the limitations of their research and indicated directions for future research. The conclusions presented at the end of the article are fully based on the conducted analyses. In general, I have no comments on the substantive part. Below are some editorial notes that authors should take into account when preparing the final version of the article.

Line 51 - move the 2 in CO2 to the subscript. In addition, it is difficult to compare emissions in different units. Here it is as CO2, and above it in C.

Figure 1 - please explain what the right part of the figure shows.

Line 90 and 92-93 - this looks like a repetition of the same content.

Lines 201 and 202 - please standardize the notation of the area. Either 2ha without space, or 2 ha.

Lines 259-261 - are these capital letters necessary?

Lines 278 and 308 - please unify the references to figures.

Lines 330 and 333 – 1,510 or 1510?

Line 375 - 8-14 times. Higher or lower? The reader can guess this, but please elaborate.

Line 325 - probably should be number 4.1. Consistently, the numbering throughout the chapter needs to be corrected.
